# Structure of Strongly Adsorbed Polymer Systems: A Computer Simulation Study

**DOI:** 10.3390/ma16175755

**Published:** 2023-08-23

**Authors:** Patrycja Olczyk, Andrzej Sikorski

**Affiliations:** 1Faculty of Chemistry, Warsaw University of Technology, Noakowskiego 3, 00-664 Warsaw, Poland; 2Faculty of Chemistry, University of Warsaw, Pasteura 1, 02-093 Warsaw, Poland

**Keywords:** lattice model, polymer film, polymer collapse, polymer melt, Monte Carlo method

## Abstract

The structure of very thin polymer films formed by strongly adsorbed macromolecules was studied by computer simulation. A coarse-grained model of strictly two-dimensional polymer systems was built, and its properties determined by an efficient Monte Carlo simulation algorithm. Properties of the model system were determined by means of Monte Carlo simulations with a sampling algorithm that combines Verdier–Stockmayer, pivot and reputation moves. The effects of temperature, chain length and polymer concentration on the macromolecular structure were investigated. It was shown that at low temperatures, the chain size increases with the concentration, that is, inversely with high temperatures. This behavior should be explained by the influence of inter-chain interactions.

## 1. Introduction

The physics of polymers is relatively well known for polymers in solution; however, the issue of polymers at the surface, e.g., polymer adsorption, is not well understood yet in spite of the fact that the behavior of such systems has been attracting considerable interest in recent years [1,2,3,4,5,6,7]. Investigation of polymer ultrathin films has also recently become one of the most interesting directions in material sciences because of potential industrial applications like colloidal stabilization, lubrication, adhesion, surface coating, wetting, etc. [8]. The structure of adsorbed polymers has been studied experimentally using quasi-elastic light scattering, atomic force microscopy, fluorescence correlation spectroscopy, or NMR methods [9,10,11,12,13,14,15,16,17,18,19,20,21]. The presence of adsorption affects properties of macromolecules compared to the polymers in solution or melt [8,22]. The case of two-dimensional polymer systems is very interesting from the theoretical point of view because strong excluded volume interactions expected here lead to behavior that cannot be observed in the three-dimensional case. Moreover, two-dimensional systems, treated for many years in polymer physics as strictly theoretical, have been obtained in a series of experiments [1,2,3,4]. Most studies refer to the weak adsorption of polymers, that is, the case of weak interactions between the surface and the polymer beads, resulting in only fragments of the macromolecule lying on the surface. A strong and irreversible adsorption case, where the polymer film is strictly two-dimensional, is also interesting because such a system can be treated as strictly two-dimensional, where the excluded volume effects are strongly revealed [23]. It is important for understanding the properties of macromolecules strongly adsorbed on surfaces, including biological systems. It may be also considered as a limiting case of ultrathin polymer films and intercalated layered silicates.

The structure of macromolecular solutions and melts in bulk and at surfaces is well known, although a dependence on both the polymer concentration and temperature is still unclear. The coil-globule transition of single polymer chains, i.e., from a state governed by entropy to a state dominated by energy, was a subject of many studies, both theoretical and experimental [24,25,26]. Two-dimensional systems were also studied, both by theory [7,22,27,28] and simulations [7,18,23,29,30,31,32,33,34,35,36,37]. The picture of such systems is more or less known, and the description is reasonably certain, although it remains debatable whether the chains are compact and segregated or are highly interpenetrated [7,18,32]. Theories and computer simulations proved that the collapse transition in two dimensions is of second order [33,38,39,40,41,42,43,44,45,46,47]. Longer chains in sufficiently poor solvent appeared to collapse through a crumpled globule to an equilibrium globule [48], and the molten globule state was also found for two-dimensional systems [49].

The question arises as to how the chains in dense polymer systems behave when an additional factor of temperature comes into play. For this purpose, we studied in this work the properties of strictly two-dimensional polymeric systems. The model included a coarse-grained chain using a lattice approximation (a square lattice). Many time and length scales in the polymer systems lead to computational problems, and the usage of simplified coarse-grained models is one of the possible solutions [50,51]. An efficient dynamic Monte Carlo simulation algorithm, which is a proper combination of commonly used sampling procedures, was employed in order to determine properties of the model studied. We focused our research on the interplay between the polymer concentration and temperature (or solvent quality) in influencing the size of chains and structure of the polymer layer.

## 2. Model and Method

The present work focuses on the theoretical determination of the structure of polymer-adsorbed systems using Monte Carlo techniques. A coarse-grained model of flexible and linear macromolecules was studied. It was assumed that all macromolecules in the system were strongly adsorbed, which meant that they were strictly two-dimensional. In other words, we studied a monolayer formed of polymer chains. The model chains consisted of a sequence of identical segments, and one segment could represent a longer real chain fragment. This model contains fewer details when compared to coarse models of polymers where united atoms consist of 5–10 real atoms [50]. In our model, a polymer segment corresponds to 10–20 chemical mers. Space discretization was introduced, which drastically reduced the number of conformations: the use of a lattice model makes it possible to increase the efficiency of computer simulation by speeding up calculations by several orders of magnitude. Therefore, polymer segments were placed in vertices of a two-dimensional square lattice. The lattice used, with a coordination number of 4, is spanned by network vectors: [0, 1], [0,−1], [1, 0], [−1, 0]. Polymer segments interacted with a square-well potential defined as:(1)$$V_{ij}\left(r\right)=\left\{\begin{array}{c} +\infty \;for\;r_{ij}<1\\ \xi \;for\;r_{ij}=1\\ 0\;for\;r_{ij}<1 \end{array}\right.$$
where *ξ* = *ε/kT*, ε is a contact energy of polymer segments, *k* = 1 is the Boltzmann constant, and *r_ij_* is a distance between a pair of polymer segments. This potential could be treated as semi-empirical, contrary to more sophisticated potentials used in mentioned coarse-grained models [50] where united atoms interact with potential of mean force [50]. The inverse of the parameter *ξ* can be used as a measure of temperature: the reduced temperature *T** = 1*/ξ* = *kT/ε*.

The total energy of the system depends on the length, number of chains and polymer concentration, and it is given in the following simple formula:(2)E/kT=ξ·ν 
where *ν* is the total number of polymer segment–polymer segment contacts in the system. In this study, a combination of dynamic Monte Carlo algorithms was used, in a variant for two dimensions, because a single algorithm did not allow simulation of such a complex system [36,37]. The following algorithms were used:(i)Reptation, which was originally connected with the motion of a macromolecule in a dense polymer liquid limited by the entanglements of other chains. The algorithm itself, which mimics the above movement, involves adding a segment to one end of the chain and truncating it from the other end. This algorithm is not ergodic, as a situation can happen where the ends of the chain do not have free lattice nodes around them on the lattice. This way of moving the chain is shown in Figure 1a.(ii)Pivot algorithm involves rotating a selected segment or several segments while the remaining part of the chain remains unchanged. In a square lattice, a 90°, 180°, and 270° rotation are possible. The advantage of this dynamic algorithm is that it is ergodic. The algorithm itself significantly changes the conformations of the macromolecule. For long chains or dense systems, this algorithm is not efficient, and the possibility of accepting such a move is low. The scheme of the Pivot algorithm is shown in Figure 1b.(iii)The Verdier–Stockmayer algorithm allows local changes in the conformation of the chain. The mechanism of this algorithm is to change the position of one or two segments that are adjacent to each other along the chains (taking into account the preservation of the allowed geometry of the chain). The chance of accepting such a move is rather high. The disadvantage of this algorithm is that it is not ergodic in two-dimensional systems and its efficiency decreases significantly in dense systems and as the temperature decreases. Representative moves in the Verdier–Stockmayer algorithm for a square lattice are shown in Figure 1c.

## 3. Results and Discussion

The model uses periodic boundary conditions in the x and y directions, which make it possible to treat a small system as infinite, albeit with an artificial periodicity property. They also make it possible to keep the density constant in an open system. The systems studied were put into a Monte Carlo cell of size *L* × *L* = 1000 × 1000. The cell size is much larger than the average diameter of the chain: L >> 2<*R*_g_^2^>^1/2^, where <*R*_g_^2^> is the mean-squared radius of gyration of the polymer chain. Simulations were carried out for chains of lengths between *N* = 10 and 100. Another variable that was introduced was the temperature, which was varied from *T** = 1 to *T** = 10. This selection of the temperature range was based on our previous studies of similar models. An athermal system characterized by the absence of attracting interactions, what corresponded to the case *T** = ∞, was also investigated. The concentration *φ* can be defined as:(3)$$\varphi =\frac{N\cdot n}{L^{2}}$$
where *n* is the number of chains in the Monte Carlo cell. Systems with different polymer concentrations varying between *φ* = 0.1 and 0.7 were studied. Concentrations of 0.1 and 0.2 indicate dilute systems, while 0.5 to 0.7 correspond to concentrated polymer solutions. Interactions between polymer segments assumed *ε* = −1 based on the previous findings. In the system, there are no explicit solvent molecules, and therefore, polymer–solvent and solvent–solvent interactions were assumed to take the value 0. A single simulation run for a given set of parameters consisted of 10^7^–10^9^ Monte Carlo steps, where a step consisted of trying to move each of the polymer segments in the system using each type of local move. Longer simulations (10^9^ steps) were performed for low temperatures, where we encountered poor algorithm performance, and for long chains, which were characterized by a significantly longer relaxation time.

Simulations were performed to study various structural properties of the system. The parameters that described the size of the chain were the mean-squared radius of gyration <*R_g_*^2^> and the mean-squared end-to-end distance <*R*^2^> averaged values over all chains. Figure 2 shows the results of system simulations on the mean-squared end-to-end distance <*R*^2^> with the change of temperature for some polymer concentrations. The higher the temperatures, the higher the <*R*^2^> values. At higher temperatures, the concentration causes the polymer chains to compress, as attraction interactions play an important role at this point and the chains collapse. In addition, the lower the polymer concentration, the greater the increase in <*R*^2^> at higher temperatures, while for low temperatures the relationship is reversed. This means that both concentration and temperature have a significant effect on the size of the polymer chains. For long chains of *N* = 100, it was observed that high concentrations with increasing temperature result in the highest <*R*^2^> values compared to shorter chains. In this case, the highest <*R*^2^> values were obtained by systems with the lowest concentrations with increasing temperature. The greatest decrease in size of polymer chains with temperature occurs for the smallest values of *φ*. The region of sharp decrease in polymer size for *N* = 10, 25 and 50 is visible at a temperature around *T** = 2. However, for long chains of *N* = 100, it can be seen that this value shifts towards higher temperatures.

The dependence of the mean-squared radius of gyration <*R_g_*^2^> on temperature T was also investigated, since this parameter is more accurate and precise than <*R*^2^>. This dependence is shown in Figure 3. It can be seen that the curves take a similar shape to the graphs of <*R*^2^>. As the temperature increases, the mean-squared radius of gyration of the polymer chains increases. In addition, the largest increase occurs between the transition from temperature *T** = 1 to *T** = 2. As with the previous graphs of <*R*^2^>, it was noted that the inflection point for long chains shifts toward higher temperatures compared to shorter chains.

The ratio of the two size parameters discussed above is a good indicator of the effect of temperature [41]. Figure 4 shows a plot of the <*R_g_*^2^>/<*R*^2^> ratio as a function of temperature for some polymer concentrations. As the temperature increases, this ratio decreases in all cases considered and no peak is seen, the presence of which would indicate some sort of transition. It can be concluded that as the temperature increases, the polymer coils become more packed finally reaching the state of a densely packed globule and the interactions between the chains decrease. The ratio <*R_g_*^2^>/<*R*^2^> is lower for high temperatures and increases sharply at lower temperatures as the polymer chains collapse. For two-dimensional polymer systems with excluded volume, the value of this ratio predicted by the renormalization group theory is 0.133 [48], while without excluded volume, it takes the value of 0.167 [49]. Computer simulations of long chains on a square lattice have yielded a value for this ratio of 0.140 [50]. High temperature values presented in Figure 4 are considerably higher than these previous findings. However, it should be stressed that all of the values mentioned above (0.133 and 0.140) were predicted for a single chain, while there are no theoretical predictions for dense many-chain systems. The higher the polymer concentration, the higher the value of this ratio. Lowering the temperature leads to the increase in this ratio and to an increase in the difference between simulation results and predictions for a single chain, and interestingly, this increase is greatest for the most dilute solutions under consideration (*φ* = 0.1).

Figure 5 shows plots of the mean-squared radius of gyration <*R_g_*^2^> as a function of polymer concentration *φ* for different chain lengths with different temperatures. The athermal systems with *T** = ∞ were included for comparison. The size of chains decreases with the increase in the polymer concentration (for *T** = 2, the decrease is negligible) for athermal systems and at temperatures below the transition from coil to globule. For low temperature, the situation is reversed: as the concentration increases, lower <*R_g_*^2^> values are seen. These trends are present for polymer systems consisting of shorter chains, i.e., for *N* = 10, 25 and 50. With long chains of *N* = 100, the size of the macromolecule increases with increasing polymer concentration. For higher temperatures (*T** = 3.3 and athermal systems), a decrease in <*R_g_*^2^> values can be seen because chains simply become compressed. In contrast, at lower temperatures (*T** = 1, and 2), chain contraction is promoted by chain collapse due to segment–segment interactions, and on the other hand, some of these interactions are inter-chain and hinder the formation of compact disks. This is particularly evident for the low temperature *T** = 1, where the increase in the polymer concentration leads to an increase in the average chain size. De Gennes scaling considerations for systems without an excluded volume lead to an inversely linear dependence of macromolecule size on the total polymer concentration <*R_g_*^2^>~*φ*^−1^ [22]. In the case of our model, it can be noted that this relation does not work even for athermal systems where the dependency is close to linear. One has to remember that the introduction of the excluded volume has to change the scaling as well as polymer–polymer attractive interactions. Moreover, the total density used in the de Gennes formula is, therefore, different than that in clusters formed at low temperature by collapsed chains, where polymer density is considerably higher. We worked with dilute, semi-dilute, and at the onset of dense systems. The percolation point that can be roughly identified with the chain overlap point is located between *c_p_* = 0.50 and 0.34 for an athermal single chain consisting of *N* = 10 and 100 beads, respectively. The introduction of the temperature makes the percolation point independent of N: *c_p_* = 0.40 for *T* = 3.3 to 0.40 for *T* = 1 [52]. It should be remembered that at low temperatures, the density of the system is not uniform. One can try to estimate it by the local density, i.e., the ratio *N*/<*R_g_*^2^> [53]. For the chain *N* = 100, one can discuss the behavior of local density, but the chain *N* = 10 is very short, and it is not exactly a coil or a disk, so considering local density does not really make sense here. The introduction of the attractive energy into the diluted system (*φ* = 0.1) increases *N*/<*R_g_*^2^> significantly: from ca. 2.1 to ca. 4.9 for an athermal system and *T** = 1, respectively. The other estimation of the polymer concentration in low-temperature clusters gives the value 0.91. Increasing the polymer concentration at higher temperatures interferes with the formation of discrete disks, and the local density increases. At low temperatures, the effect of concentration is exactly the opposite, i.e., the local density decreases.

The total energy of the system, denoted as *E* calculated according to Equation (2), was also examined. It is difficult to compare the energy at different temperatures, chain lengths and numbers of these chains in the system, so it was decided to use instead a number of contacts denoted by *ν*. The number of contacts includes both contacts of a segment with segments from other chains and contacts with segments from the same chain summed up after all segments in the system. The maximum number of contacts *ν_max_* was also estimated for each system, and the reduced values *ν*/*ν_max_* were considered. The reduced number of contacts was then analyzed as a function of temperature, as shown in Figure 6. The length of the chains does not significantly affect the shape of the curves. The lower the temperature, the more contacts occur, because polymer–polymer contacts are preferred, and the system then has a lower energy. An increase in the reduced number of contacts is also visible where the polymer concentration increases, and for low temperatures, the number of contacts increases rapidly.

Another variable that was analyzed was the reduced number of contacts in the system as a function of polymer concentration presented in Figure 7. The reduced number of contacts increases with the increase in polymer concentration. This number also increases with chain length, especially at low polymer concentration. For polymer concentrations between *φ* = 0.3 and 0.4, large fluctuations in the number of contacts were observed for both chain lengths. They are significant for intermediate temperatures. This non-monotonic behavior of the number of contacts, i.e., of the total energy, can be attributed to the competition in the formation of intra-chain and inter-chain energetic contacts: at these temperatures, the chain collapse (the maximizing of intra-chain contacts) is being disturbed by the presence of inter-chain contacts. It should be noted that these fluctuations occur for concentrations that are located close to the percolation threshold, i.e., where chains begin to touch each other, and hence, the number of inter-chain contacts increases. What is more, at intermediate temperatures, polymer–polymer contacts are quite weak and easy to break.

The next analysis concerned the dependence of chain size on chain length. Figure 8 shows the change in the mean-squared radius of gyration with chain length for different concentrations at two temperatures: *T** = ∞ (athermal system) and *T** = 1 (low temperature). The dependence of <*R_g_*^2^> on N was shown on a log–log scale because scaling behavior of the type <*R_g_*^2^>~*N*^2ν^ was expected. Theoretical predictions for diluted (the exponent 3/2) and concentrated (the exponent 1) systems were also marked in this Figure. The behavior of the shortest chains deviates from the others, but it must be remembered that scaling laws were derived for infinitely long chains. After discarding the shortest ones, the others clearly satisfy the expected relationship. One can observe that for the lowest polymer concentration studied (*φ* = 0.1), the exponent is considerably lower than that for a diluted system. We also found that in the athermal system, the exponent 2*ν* decreases from 1.14 ± 0.01 to 1.01 ± 0.02 when the polymer concentration increases from 0.1 to 0.7, which was close to the value obtained for the same model but using a different sampling algorithm and at higher densities where 2*ν* was found between 1.08 and 1.1 [54], while off-lattice simulations gave the exponent between 1.51 and 1.07 [55]. This means that it reaches the value of 1 predicted theoretically for concentrated polymer melts [56]. This behavior is understandable, because when the polymer concentration increases, the chains begin to feel the presence of others, shrink and interpenetrate. In the low temperature system, the exponent behaves inversely, that is, it increases from 0.77 ± 0.02 to 0.93 ± 0.02 when the polymer concentration increases from 0.1 to 0.7. Note that several processes are taking place here at once. First, the chains must shrink with concentration, as in an athermal system. Second, attractive intra-chain interactions lead to collapse and the formation of densely packed disks (the size of such disks should scale like *N*^1^). Third, at higher densities, inter-chain interactions begin to have an impact, which interferes with the formation of compact disks.

Additional insight can be obtained from the analysis of the visualization of polymer system configurations studied. Figure 9 presents examples of the system at high, intermediate and low temperatures and at intermediate density. The chain length was *N* = 50 for temperatures *T** = ∞, 2, and 1. The visualizations show the polymer chains, which have been separated by color so that they can be distinguished from each other. At high temperatures (Figure 9a), all contacts are random and the chains are coiled, some of them have no contact with other chains at all, and there is no chain–chain interpenetration observed. At intermediate temperatures (Figure 9b), the chains collapse and form irregular disks, although some chains are only partially folded. Most of the chains are in contact with other chains, while the number of chains folded into disks and separated from other macromolecules is rather limited. There are chains that still remain stretched, having contacts only with other chains. At low temperatures (Figure 9c), the chain folding process has progressed further, and almost all of them are in the form of compact disks. Moreover, chains become tightly packed, forming some clusters. Small chain–chain interpenetrations can be seen. Similar interpenetration was found in the case of dense two-dimensional athermal melts [57,58,59] and in AFM experiments [41]. Changes in the local polymer density can also be easily seen in this figure.

## 4. Conclusions

A theoretical study concerning polymer systems strongly adsorbed on a homogeneous surface was carried out. A simple model of two-dimensional polymer systems was designed using a coarse-grained representation and lattice approximation. Static properties of these systems were determined by means of computer simulations employing the Monte Carlo method. We focused on studies on the influence of parameters like chain length, polymer concentration and temperature on the structure of the polymer film.

Analysis of simulations results of coarse-grained models under the conditions specified above allowed the following general conclusions to be drawn. At high temperatures, chain sizes decrease with polymer concentration, which was already expected from theoretical considerations. The lower the temperature, the smaller the size of the polymer chains, due to their higher packing. Moreover, at low temperatures, the chains’ behavior starts to be opposite when compared to high temperatures: the size of a macromolecule increases with concentration. At low temperatures, intra-chain interactions are strong, but as the concentration increases, polymer chain sizes increase, the number of inter-chain contacts increases, and thus, the chain–chain interpenetration is visible.

## Figures and Tables

**Figure 1 materials-16-05755-f001:**
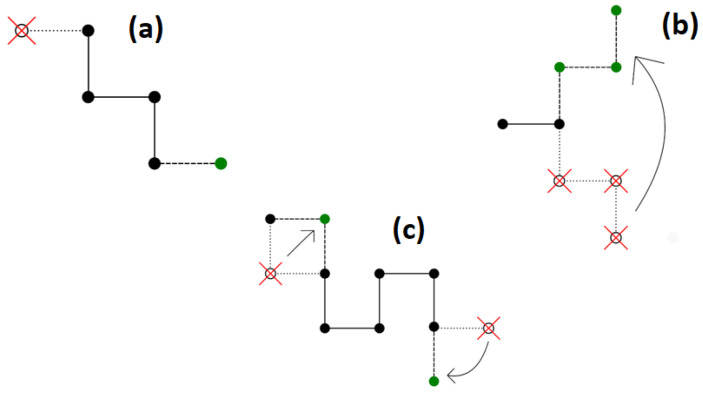
A scheme for modifying the conformation of a chain using the algorithms of: (**a**) reptation; (**b**) pivot; (**c**) Verdier–Stockmayer. Old positions of chain beads are marked by open circles and red crosses, while new positions are marked in green.

**Figure 2 materials-16-05755-f002:**
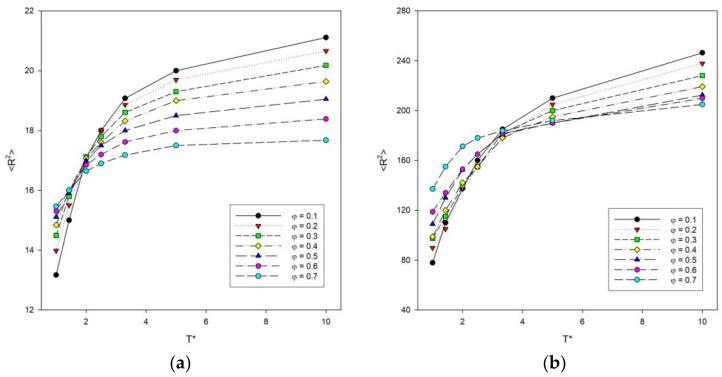
Plots of the mean-squared end-to-end distance <*R*^2^> vs. temperature *T** for chain length *N* = 10 (**a**) and *N* = 100 (**b**). Polymer concentrations *φ* are given in insets.

**Figure 3 materials-16-05755-f003:**
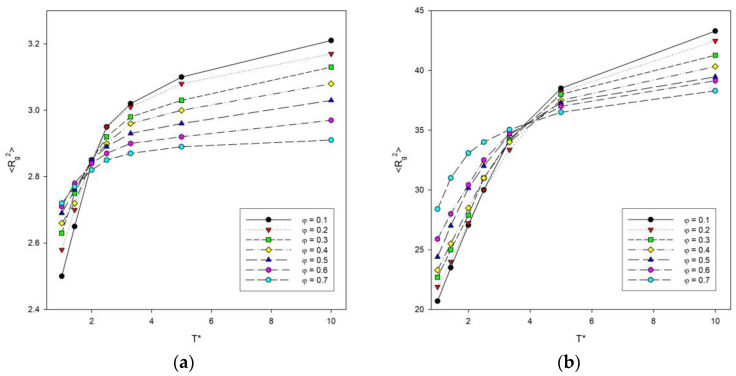
Plots of the mean-squared radius of gyration <*R_g_*^2^> vs. temperature *T** for chain length *N* = 10 (**a**) and *N* = 100 (**b**). Polymer concentrations *φ* are given in insets.

**Figure 4 materials-16-05755-f004:**
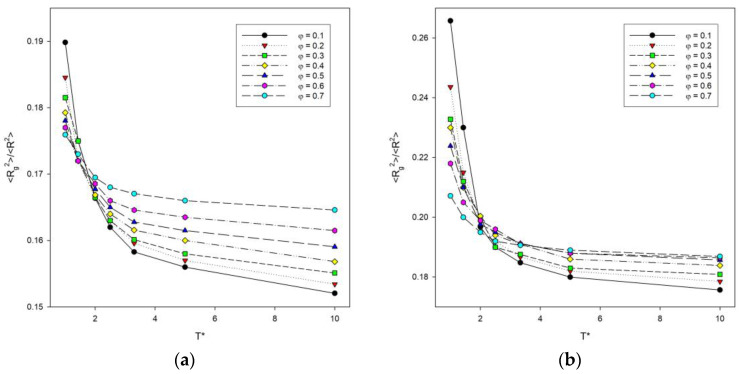
Plots of the ratio of the mean-squared radius of gyration to the mean-squared end-to-end distance <*R_g_*^2^>/<*R*^2^> vs. temperature *T** for chain length *N* = 10 (**a**) and *N* = 100 (**b**). Polymer concentrations *φ* are given in insets.

**Figure 5 materials-16-05755-f005:**
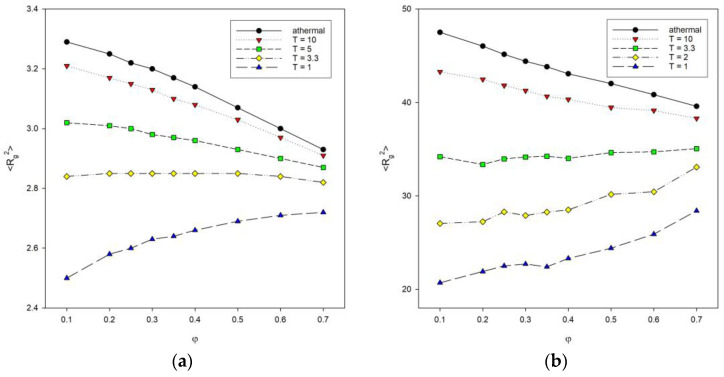
Plots of the mean-squared radius of gyration <*R_g_*^2^> vs. polymer concentration *φ* for chain length *N* = 10 (**a**) and *N* = 100 (**b**). Temperatures are given in insets.

**Figure 6 materials-16-05755-f006:**
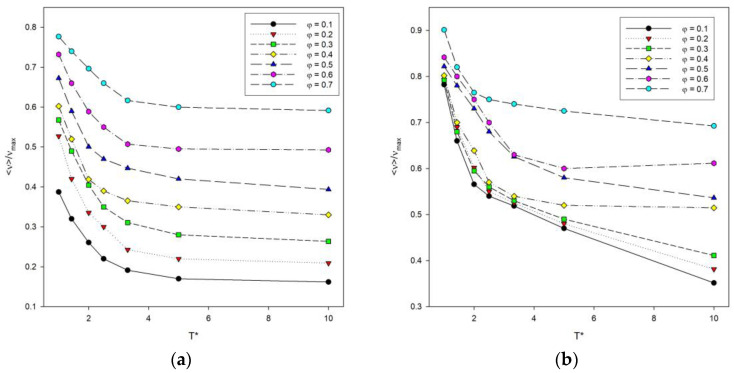
Plots of the reduced number of polymer–polymer contacts <*ν*>/*ν_max_* vs. temperature *T** for chain length *N* = 10 (**a**) and *N* = 100 (**b**). Polymer concentrations *φ* are given in insets.

**Figure 7 materials-16-05755-f007:**
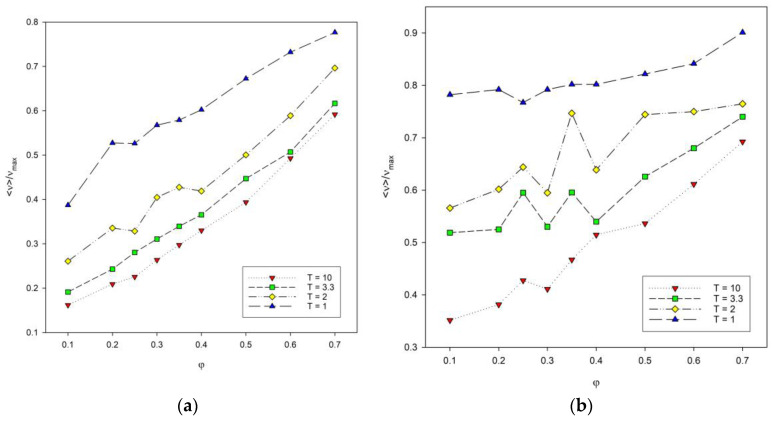
Plots of the reduced number of polymer–polymer contacts <*ν*>/*ν_max_* vs. polymer concentration *φ* for chain length *N* = 10 (**a**) and N = 100 (**b**). Temperatures are given in insets.

**Figure 8 materials-16-05755-f008:**
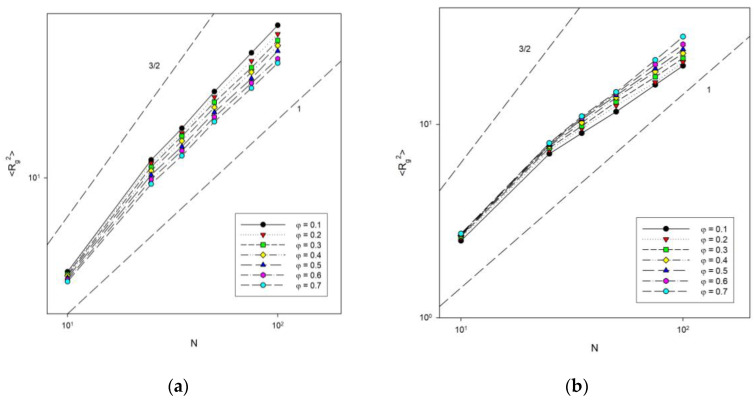
Plots of the mean-squared radius of gyration vs. chain length for *T** = ∞ (**a**) and *T** = 1 (**b**). Polymer concentrations *φ* are given in insets.

**Figure 9 materials-16-05755-f009:**
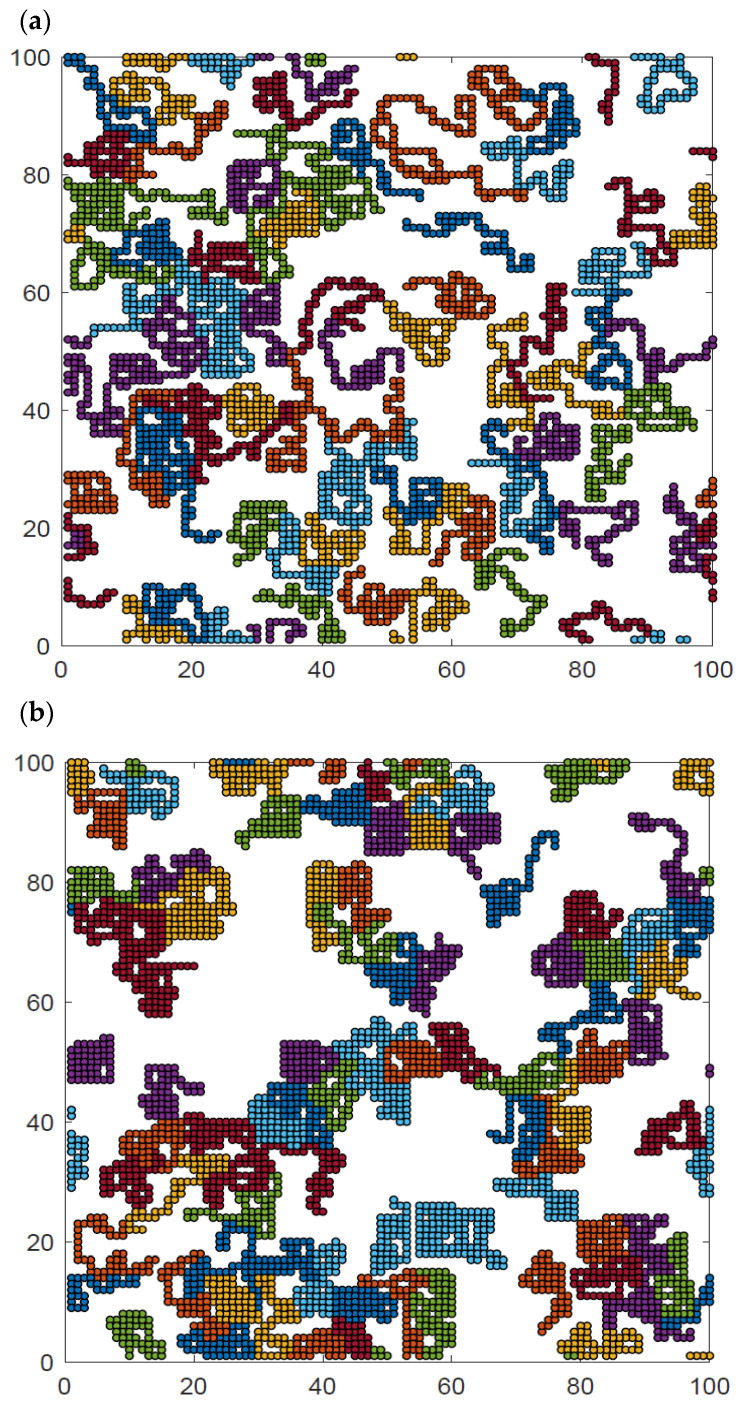
Snapshots of the system *N* = 50, *L* = 100 and *φ* = 0.4 for temperatures *T** = ∞ (**a**), *T** = 2 (**b**), and *T** = 1 (**c**). Each chain is marked with a different color.

## Data Availability

The data that support the findings of this study are available from the corresponding author upon reasonable request.

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
