# Peer review of "Structure of Strongly Adsorbed Polymer Systems: A Computer Simulation Study"

_materials, 2023, doi:10.3390/ma16175755_

Round 1
Reviewer 1 Report
The paper Manuscript Number materials-2495381 reported coarse-particle size model was used to study the strong adsorption polymer film system by Monte Carlo simulation, and the effects of chain length, polymer concentration and temperature on the polymer film structure were mainly studied。In my opinion, the quality of manuscript may improve before publication in Materials. Following are some specific comments:
1. The article has been emphasizing the strong adsorption polymer system, may I ask how this strong adsorption is reflected in the modeling process? How can we prove the reliability and rationality of the model?
2. What is the principle of choosing a simulated temperature range of 1 to 10? Is there any special significance for strongly adsorbed polymer film materials?
3. Figure 7 (a) and (b) are comparison graphs. It is suggested that the shape and color of the dot plot should be the same for the same temperature. The description of this figure is not clear, such as the description in the article” For concentrations between j = 0.3 and 0.4, large fluctuations in the number of contacts was observed for each chain length for different temperatures.” But it does not analyze the specific reasons causing this change.
4. What is the relationship between this paper and strong adsorption? Through the simulation data of the article seems difficult to reflect, the article focuses on the study of the change the mean square of the radius of gyration <Rg 2>, in fact more parameters that could characterize adsorption were not studied
I can recommend this paper for publication in Materials on the basis reasons described above: My recommendation for the submission is Major revision.
Author Response
The answers are attached.

Reviewer 2 Report
In this manuscript Olczyk and Sikorski have done MC simulations to investigate the structure of strongly adsorbed polymer chains. Mainly, they have studied the influence of temperature on the structure of polymer. I have a few concerns, to be addressed before I can recommend publication of this polymer.
1-In most applications, as addressed by authors in the introduction, the polymer chains are adsorbed on a substrate. Considering this point, what is the relevance of the present study, in which no substrate explicitly exists in the model, to the majority of Engineering applications?
2-It is known that the coarse-grained beads interact through much softer potentials than the individual atoms (whose potential energy of interactions is mostly specified as a Lennard-Jones potential). Is there any specific reason the authors have used a hard-core potential (square-well), which does not seem so realistic for their system?
3-In the literature, there are realistic coarse-grained models of long polymer chains adsorbed on a realistic substrate (see for example J. Phys. Chem. C 2013, 117, 5249). To show the relevance of their work to adsorbed polymers on a substrate, I would like to ask the authors to compare their results (at least some of their structural results) with those.
4-Page 4, the sentence “Polymer-solvent and solvent-solvent interactions were assumed to take the value 0.” is misleading, because in this work no explicit solvent is taken into account in the model. I am not even sure if the role of solvent is taken into account implicitly. Even in that case, the authors need to clearly mention it in the text.
5-Some minor points: “mean square of the radius of gyration” → “mean-squared radius of gyration” and “mean square of the distance between chain ends” --→ “mean-squared end-to-end distance” or “mean-squared-distance between the chain ends”.
In this manuscript Olczyk and Sikorski have done MC simulations to investigate the structure of strongly adsorbed polymer chains. Mainly, they have studied the influence of temperature on the structure of polymer. I have a few concerns, to be addressed before I can recommend publication of this polymer.
1-In most applications, as addressed by authors in the introduction, the polymer chains are adsorbed on a substrate. Considering this point, what is the relevance of the present study, in which no substrate explicitly exists in the model, to the majority of Engineering applications?
2-It is known that the coarse-grained beads interact through much softer potentials than the individual atoms (whose potential energy of interactions is mostly specified as a Lennard-Jones potential). Is there any specific reason the authors have used a hard-core potential (square-well), which does not seem so realistic for their system?
3-In the literature, there are realistic coarse-grained models of long polymer chains adsorbed on a realistic substrate (see for example J. Phys. Chem. C 2013, 117, 5249). To show the relevance of their work to adsorbed polymers on a substrate, I would like to ask the authors to compare their results (at least some of their structural results) with those.
4-Page 4, the sentence “Polymer-solvent and solvent-solvent interactions were assumed to take the value 0.” is misleading, because in this work no explicit solvent is taken into account in the model. I am not even sure if the role of solvent is taken into account implicitly. Even in that case, the authors need to clearly mention it in the text.
5-Some minor points: “mean square of the radius of gyration” → “mean-squared radius of gyration” and “mean square of the distance between chain ends” --→ “mean-squared end-to-end distance” or “mean-squared-distance between the chain ends”.
Author Response
The answers are attached.

Reviewer 3 Report
Understanding the behavior of 2D polymers with attractive interactions is an active area of research, with implications for a wide range of applications in materials science, biophysics, and nanotechnology.
The authors aim to investigate scaling of strongly adsorbed polymers at various temperatures and concentrations using Monte Carlo simulations in Silico as they claim: "The structure of macromolecular solutions and melts in bulk and at surfaces is well known although a dependence on both the polymer concentration and temperature is still unclear."
Much of this study is consistent with the findings of the existing literature and may provide information that complements the findings of the existing literature.
The new results relate to the role of "imposed" concentrations and should be presented in a comprehensive form.
1. The current work is dedicated to measurement of polymer size with various "imposed" concentrations. A theoretical model should be relevant to a physical system.
There seems to be a limit to the concentration at which strongly adsorbed polymers can exist in two-dimensional solution without overlapping each other.
What are the physically realistic values of concentration and surface energy that satisfies this condition?
2. At low temperature, one can expect that polymers undergo phase separation to dense phase and supernatant. When the given imposed density is small, say \phi < 0.5, the surface is not uniformly covered and there are large fluctuations.
The concentration of dense phase would be different from the imposed concentration.
It seems more plausible to consider the concentration of the dense phase for comparison with theoretical predictions.
For all imposed concentrations, average chain sizes increase with increasing temperatures. There is no surprise. On the other hand, authors reported the concentration dependence of the average chain size is reversed at low temperatures. The local density of collapsed part should be taken into accounts. Authors should identify the critical temperature of collapse transitions and rationalize this finding.
At dense phase, one expects that the exclude volume effect would be screened at large scales. When the imposed concentration is not large enough, the dense phase is only local, it seems that the screening is not complete even below the critical temperature.
It is also unclear to me whether the system is equilibrated. How does the domain size grow with the interaction strength (1/T) below the transition temperature?
3. The authors reported the tendency of chain size growth as a function of the imposed concentration at low temperature (below the temperature of collapse transition).
Scaling theory predicts that polymers in melts are locally swollen in the scale of the concentration blob (due to the excluded volume) and recover R^2 ~ N in large scale in 2D. This behavior is strictly different from ideal Gaussian chain. Authors did not exploit the intra strand scaling but the scaling of chain size shows the cross over from N^{3/2} to N^{1}, consistent with theoretical predictions.
The dependence on the imposed concentration is less obvious.
Although the scaling theory is well known, for the understanding of the reader, the authors should summarize the relevant scaling predictions especially when the interaction between monomers is attractive (for the case that the second virial coefficient is negative).
4. There are some errors in using of Greek font (e.g. expressions Eq.1,2, and 3).
There are typos in Eq. 1,2, and 3 and also in line 188.
Author Response
The answers are attached.

Round 2
Reviewer 2 Report
The revised version of manuscript is improved considerably. I recommend publication.
Author Response
The authors thank the reviewer.
Reviewer 3 Report
Report on materials 2495381 (2nd Round)
The paper report MC simulation results for a 2D monolayer system.
The manuscript mainly reports the average size of polymers (oligomers) at various "imposed concentrations" and "temperatures".
I my opinion, the findings can be interpreted based on the existing theories and the paper can be further improved in the analysis.
I have requested a major revision to check a few points.
It seems that my request was not clear enough.
1) As the authors use lattice model, obviously it is difficult to address the density of local structure directly. Therefore, it is useful to demonstrate the intra chain correlations to better identify the chain structure and related scaling law from longer chain simulations.
The intra strand correlation would provide the length scales where excluded volume influences before it is screened asymptotically. The temperature dependency can be found in the length scale of the crossover.
2) The imposed concentration should be distinguished from the chain concentration of the dense phase.
Because the system is not uniform, especially at low temperatures, the concentration used in scaling relation is very different from the imposed concentration "\phi".
On line 225, authors wrote that "<R_g^2> ~ \phi ^{-1} does not work", presumably because the comparison is given with the imposed density. This statement can be misleading.
I understand the phase behavior can be subject of a separate paper, but dense phase concentration should be identified for an appropriate analysis.
3) In response to my question concerning large fluctuation, the authors claim that the large fluctuation is due to the competition between inter chain and intra chain energetic contacts.
(onlines 252-257 " For polymer ... by the presence of inter-chain contacts.")
I am not convinced by this explanation.
In strict 2D and strong confinement, as discussed in several paper (e.g. Ref. 7 and Phys. Rev. Lett. 118, 067802), one expects chains to be segregated so that the number of inter-chain contact will be reduced.
I would like to understand the results of Fig. 6 in this context.
(I assume that "segment-segment contact" means contacts of monomers belongs to different chains.
How \nu_{max} is evaluated is still unclear to me.)
Large fluctuations may be the signature of a phase transition to dense phase.
Author Response
The authors thank the reviewer.

Round 3
Reviewer 3 Report
Report on materials 2495381 (3rd Round)
The authors' comment of the behavior of <R_g^2> is fine.
I would like to make it clear that the density that enters the scaling law is different from the density that is imposed.
The proper density estimation might be counting the number of monomers in an island. For example, from Fig. 8 (c), authors may estimate density of low right cluster.
Author Response
All questions were answered.
